# Vision-Based System for Automated Estimation of the Frontal Area of Swimmers: Towards the Determination of the Instant Active Drag: A Pilot Study

**DOI:** 10.3390/s22030955

**Published:** 2022-01-26

**Authors:** José M. González-Ravé, Francisco Moya-Fernández, Francisco Hermosilla-Perona, Fernando J. Castillo-García

**Affiliations:** 1Sport Training Laboratory, Faculty of Sports Sciences, University of Castilla-La Mancha, 45071 Toledo, Spain; Francisco.Hermosilla@uclm.es; 2School of Industrial and Aerospace Engineering of Toledo, University of Castilla-La Mancha, 45071 Toledo, Spain; Francisco.Moya@uclm.es; 3Facultad de Ciencias de la Vida y la Naturaleza, Universidad Nebrija, 28248 Madrid, Spain; 4Department of Physical Activity and Sports Science, Alfonso X El Sabio University, 28691 Madrid, Spain

**Keywords:** swimming, active drag, resistive force, computer vision, area determination

## Abstract

Swimmers take great advantage by reducing the drag forces either in passive or active conditions. The purpose of this work is to determine the frontal area of swimmers by means of an automated vision system. The proposed algorithm is automated and also allows to determine lateral pose of the swimmer for training purposes. In this way, a step towards the determination of the instantaneous active drag is reached that could be obtained by correlating the effective frontal area of the swimmer to the velocity. This article shows a novel algorithm for estimating the frontal and lateral area in comparison with other models. The computing time allows to obtain a reasonable online representation of the results. The development of an automated method to obtain the frontal surface area during swimming increases the knowledge of the temporal fluctuation of the frontal surface area in swimming. It would allow the best monitoring of a swimmer in their swimming training sessions. Further works will present the complete device, which allows to track the swimmer while acquiring the images and a more realistic model of conventional active drag ones.

## 1. Introduction

### 1.1. Preliminaries

Swimming performance is affected by both the propulsive and resistive forces [1]. Regarding the hydrodynamic drag, this force can be defined as an external force that acts in the swimmer’s body parallel but in the opposite direction of his movement [2,3]. According to Newton’s second law of motion, the sum of these forces determines the acceleration of the swimmer (Equation (1)).
(1)$$m\;a=F_{P}+F_{R}$$
where *m* is the swimmer’s mass, *a* is the acceleration of the swimmer, and *F_P_* and *F_R_* are the total propulsive and resistive forces, respectively. According to Equation (1), when the magnitude of the propulsive and the resistive force are equal, a swimmer would maintain a constant speed. When, however, the amount of one force exceeds that of the other, the swimmer will accelerate or decelerate. Thus, swimmers should have an ability to both produce large propulsive force and to reduce resistive forces for the purpose of improving performance. Quantifying these forces when swimming is therefore very beneficial in improving our understanding about swimming technique and improving swimming performance. In this sense, it has been found that speed in swimming sprint events is positively correlated to maximal force and mean force production [4,5]. Nevertheless, an increase in force production is not directly linked with an increment in swimming velocity because swimming drag could be also higher [6]. For instance, expert swimmers present higher propulsive forces than less expert swimmers but do not show higher drag values, therefore obtaining higher swimming speed and performance [5].

Because there are currently no methods that allow researchers to quantify these forces directly, several methods for indirect estimation of propulsive and resistive forces have been developed to assess either the mean forces over a period of time or time-series force data. Although mean propulsive/resistive force values would provide an overview of the forces related to swimming strokes or swimming skill levels, a better insight on technique can be gained with a time-series approach. For example, in the last decade, time-series propulsive forces produced by the hands or feet have been quantified with the use of pressure sensors [7,8,9,10], which has provided insight into relationships between swimmer’s motion and propulsive forces. Examples of such knowledge include the positive effect of shoulder roll angular velocity on the propulsive force production in front craw [7,8], differences in the time-series leg propulsive force between an elite and sub-elite breaststroker [9], and the increase in the propulsive impulse (despite the decrease in the stroke cycle duration) when increasing the front crawl swimming velocity [10].

On the other hand, time-series resistive force information has only been scarcely explored in the literature even though the mean resistive force over a period of time has been investigated using different methods. One of the first methods developed to estimate resistive forces was the Measuring Active Drag (MAD) system [3,11]. This method quantifies the mean resistive force based on a directly measured reaction force from pushing-pads, which are placed in fixed depth and between-pad distance. A limitation of this method, though, is that it only allows calculation of resistive forces in arm-only front crawl swimming and also that the propulsive mechanism (propelling forward by pushing fixed-pads of set distances and depths) is quite different to the one used in actual front crawl swimming. In the 1990s, an alternative method (Velocity Perturbation Method (VPM)) was proposed [12]. VPM is based on two maximal effort trials, one with a known additional resistance and one without, and assuming that the power outputs in the two conditions are equal, the resistive force at the maximal effort can be mathematically computed. The same calculation can also be done using free-swimming and overspeed (assisted) swimming conditions, which is known as Assisted Towing Method (ATM) [13]. In both VPM and ATM, the resistive force of all four strokes can be computed. However, disadvantages of these methods include the inability of assessing the resistive force at non-maximal efforts and potential errors due to the equal power output assumption [2,14]. More recently, the Measuring Residual Thrust (MRT) method was developed, which allows estimation of the resistive force at any speed [15]. The MRT method requires multiple trials (assumed to be performed with identical technique) in a swimming flume. This, however, presents some limitations, such as the swimming flume, which may affect the technique used and the propulsive/resistive forces, while the assumption of identical technique ignores any errors that may be caused by variation in all aspects of a swimmer’s technique. Finally, a common limitation among all the above methods is that they only provide the mean resistive force over a period of time.

### 1.2. Related Works

The most recent work of Takagi et al. [16] reviewed the literature on front crawl, focusing on propulsive and resistive forces at different swimming velocities. The relationships between energetic, biomechanical, and fluid dynamics indices in competitive swimming were established factors that determine the mean of the instantaneous magnitudes of hand velocity over some time. Experimental attempts to quantify the time-series resistive force have been scarce. The only study in which the time-series resistive force was assessed was conducted by Morais et al. [17]. The time-series resistive force was quantified by obtaining the time-series frontal surface area and forward swimming velocity and applying these to the steady-state equation below (Equation (2)):(2)FRi=12 CR Ai ρ vi2
where *F_Ri_* is the resistive force at time *i*, *C_R_* is the resistive force coefficient derived from VPM, *ρ* is the density of the water, and *A_i_* and *v_i_* are the frontal surface area and the forward velocity of the swimmer at time *i*. A limitation of this method is the application of the steady-state equation to a non-steady-state (fluctuating velocity) condition, which suggests that the absolute drag value derived from this method might not be accurate. Nevertheless, it is likely that the fluctuation of the frontal surface area affects proportionally the resistive force variation, as a larger reference area would produce a larger hydrodynamic pressure difference between the front and back sides of the body. Thus, quantifying changes in the frontal surface area during swimming should provide an insight into the resistive force fluctuation. The frontal surface area during swimming has also been assessed in another study in which intra-cycle variation of the frontal surface area was investigated in the four swimming strokes and in streamlined kicking [18]. In both studies, however, the frontal surface area was obtained at a very low sampling frequency. Morais et al. [17] obtained the surface area at only five key events in one stroke cycle and estimated the time-series one-cycle frontal surface area data using a spline function, and Gatta et al. [18] calculated the surface area with 12.5 Hz, which was 2–4 times lower than the recommended sampling frequency range for swimming motion analysis (25–50 Hz) [19].

The reason for the low sampling frequency used in the two studies was probably due to the analysis process involved. Researchers in both studies manually outlined the body in video images to obtain the frontal surface area, which was probably very time consuming. Considering that manual processing of data obtained at a high sampling frequency is both challenging and time consuming, establishing an automatic process to obtain the frontal surface area would substantially improve our ability to collect and analyse more complex and accurate data. At the moment, this is an open problem that we tried to sort out.

The purpose of the present study was to develop an automated method to obtain the frontal surface area during swimming. Such a method would be of great benefit for both swimming researchers and practitioners when exploring the relationships between technique and resistive forces in swimming, as it would lead us better understand the temporal fluctuation of the frontal surface area that is directly linked to the time-series resistive force in swimming.

## 2. Materials and Methods

### 2.1. Experimental Protocol

A set of experimental tests were conducted with one regional male swimmer (age: 20 y; body mass 68 kg; height: 173 cm; training hours: 9 h per week). The swimmer was specialized in individual medley events and had 5 years of competitive training experience. The participant visited the swimming pool in a non-fatigued state (non-intense exercise in the 48 h and no strength training in the 72 h prior to testing). The study was performed in accordance with the Declaration of Helsinki (October 2008, Seoul, Korea), and the experimental protocols were approved by the ethical committee of the local University (Approval Number UNNE-2020-010).

Before the test started, an investigator took pictures of the whole body in different positions for subsequent calculation of the dry model (see Figure 1). Following this process, the swimmer performed a 400-m standardized warm-up consisting of 200-m easy front crawl, 100 m of short sprint sets (12.5 fast and 12.5 easy), and two sets of 25-m kick and 25-m swim. All measurements were conducted on the same day in an indoor 25-m swimming pool with a water temperature of 27 degrees Celsius.

### 2.2. Hardware Description and Experimental Setting

Two underwater cameras (frontal and lateral) filmed 5 different 60-s tethered swimming trials at sub-maximal intensity with a 90-s rest between each trial. The cameras were positioned at 60-cm depth taking into account the swimming pool characteristics. The swimmer was connected to SwimOne device by means of a hardness through a steel cable [20]. For the work presented here, the drum of SwimOne was locked, and the swimmer therefore performed the swimming trials without moving forward. Figure 2a illustrates the testing setting, and Figure 2b details the camera position regarding to the swimmer, with $\theta_{L}$ and $\rho_{L}$ as the angle and distance from the reference point of the swimmer to the lateral camera and $\theta_{F}$ and $\rho_{F}$ as the angle and distance to the frontal one. The placement of the cameras was established trying to minimize $\theta_{L}$ and $\theta_{F}$ as much as possible taking into account the pool limitation. The final values that determine the cameras poses are $\theta_{F}=5.6{}^{\circ},\;\theta_{L}=11.2{}^{\circ}$, $\rho_{L}=3140$ mm, and $\rho_{F}=6352$ mm. Swimmer was placed at 3 and 6 m from the lateral and front cameras, respectively (taking as reference the swimmer hips), in order to reduce as much as possible $\theta_{F}\;\mathrm{and}\;\theta_{L}$. In this way, we could obtain an optimal balance between the validity of the measurements obtained and the quality of the swimmer image.

Both cameras were the same model and allowed to record videos of 1080 × 1920 resolution with a framerate of 30 fps (normal mode). The cameras were calibrated using a series of poles of fixed length. They were positioned at specifically known positions (sagittal and frontal plane) throughout the area in which swimmers performed each trial (Figure 3).

For illustrative purpose, Figure 4 represents two videoframes of the lateral and frontal cameras.

### 2.3. Frontal Area Detection Algorithm

Figure 5 summarizes the steps in the algorithm for automatic detection of the swimmer’s frontal area, which was developed using Computer Vision Toolbox of Matlab^TM^. Figure 5a represents the Data Flow Diagram of the Algorithm, and Figure 5b shows the equivalent pseudocode.

The algorithm starts with extracting each videoframe to be processed. These frames are captured in RGB colormap. Then, the region of interest (ROI) is selected with the purpose to decrease the computing time by analysing only the region where the swimmer is placed at the acquired image [21]. Figure 6 shows the ROI of a frontal frame for illustrative purpose.

The following steps are to detect the swimwear, the harness, and the cap of the swimmer and to replace it by a region with similar colour to the skin of the swimmer.

In this study, swimwear was dark red, and the cap was black, together with the harness of SwimOne device (see Figure 1), which were all replaced with other colours to make a clear contrast with the water colour using colour thresholding technique [22].

The resulting mask was configured in CIELAB colour space, and before this step, RGB frame was converted to CIELAB colour space, where L represents the lightness of the pixel, a  represents the value of the colour line between green and red, and b channel represents the value of the colour line between blue and yellow [23,24]. Appendix A summarizes the equations to obtain channels *L*, *a*, and *b* from RGB ones. The mask for removing swimwear in the frontal camera is defined by the following:(3)     25.466<L<98.620−46.336<a<8.860−31.446<b<60.616
with L, a, and b representing the lightness, red-green, and blue-yellow channels, respectively.

This mask was applied to identify the pixels illustrating the swimwear, cap, and harness, which were substituted in the RGB image by (RGB)=(27,131,135), which corresponded to a pixel colour of the chest of the swimmer. Figure 7 illustrates the application of the mask to a frame with the ROI of the original image.

The following steps consist of removing water area and other elements, such as lanes. These steps were implemented by other sequential masks. The first one was applied on a normalised CIELAB colourmap:(4)$$\;\;\;\;\;5<L^{*}<160$$
with $L^{*}$ as the normalization of L channel into (0, 255) range.

Other elements were removed by a filter area down to the water line, which is characterized by i=97, with i as the row coordinate of the frame ROI. The result of applying both masks is illustrated in Figure 8. The filter shown in Equation (4) together with the area filter (i≥97) also remove the possible reflexion of the image of the swimmer up to the water line.

The ROI of the video frame was then converted to grayscale and subsequently binarised to obtain the black and white image. The binarisation of the image was performed using the algorithm described elsewhere [25]. The subsequent filter area consists of removing all the blobs with an area less than a minimum allowed value (set experimentally to 3000 pixels) and to fill the resulting blob to avoid internal holes. This process is illustrated in Figure 9.

Finally, *regionprops* function was applied to the black and image to obtain the number of pixels of the remaining area [26]. Figure 10 represents the results for different frames. For illustrative purposes, the final mask was coloured in red over the original frame.

Using a laptop with a Intel(R) Core(TM) i7-8850h with 16 GB RAM and NVIDIA Quadro P2000, the computing time of the proposed algorithm is illustrated in Figure 11 for one of the experiments.

### 2.4. Extension to Lateral Area Detection

The algorithm described in Figure 5 can be also applied for detecting the lateral area of the swimmer by customizing the masks described in the frontal area determination.

Figure 12 shows the final result for lateral area estimation.

Note that both frontal and lateral area were only determined in pixels, and no conversion to the real area was performed [27]. Equation (2) is a model of the resistive force where $A_{i}$ should be expressed in m^2^ but the influence of the correlation between word coordinates and image coordinates will be a constant (f.e. one pixel $\approx 6.23\;{\mathrm{mm}}^{2}),$ and therefore, no influence over the goodness of the relationship between effective area and the linear velocity of the swimmer is expected. Nevertheless, further work will include the camera calibration procedure in order to obtain frontal and lateral areas in m^2^.

## 3. Results

### 3.1. Frontal and Lateral Area

Figure 13 and Figure 14 represent the time-series variation of the frontal and lateral area in two different experiments. The result shows the periodic tendency of the swimmer in the different cycles of the stroke.

Results shown in Figure 13 and Figure 14 should be correlated to the velocity of the swimmer to validate or modify model (2).

### 3.2. Analysis of the Results

#### 3.2.1. Correlation between Frontal and Lateral Areas

Figure 15 presents the representation of the Frontal Area vs. Lateral Area in both Test (Test I, Figure 15a, Test II, Figure 15b) together with their linear regression. Note that the coefficients of determination are very small ($R^{2}=0.1122$ and $R^{2}=0.09951),$ and therefore, no correlation between both areas is observed.

#### 3.2.2. Frequency Domain Characterization

For illustrative purpose, Figure 16 represents the frequency spectrum of frontal and lateral area for Test I.

For both measurements the frequency peaks are cle for frontal area at 0.47596 Hz and 0.47595 Hz for lateral one. Both main frequencies have similar values.

The frequency and amplitude values could be used to be correlated to the swimmer speed when the device shown in Figure 17 will be available.

## 4. Discussion

This study aimed to determine the frontal area of swimmers employing an automated vision system. The resistive forces that influence the swimmer in the water include form, wave drag (as a result of accelerating the water away from the body), and frictional drag, which are influenced by the swimmer’s velocity, boundary layer, shape, size, and the frontal surface area [28], while controversial issues exist due to the difficulty of accurately measuring the wetted surface area of the swimmer’s body [29]. We propose a vision-based System for Automated Estimation of the frontal surface area of swimmers, and these instant measurements could allow adapting the technical characteristics for swimmers into the session to improve the swimming performance reducing the passive drag. Our results showed the different time evolution of the frontal area, and the difference with the study of Morais et al. [17] is how the image in our study processes automatically, while these authors manually digitized the calculation of the frontal surface area. Morais et al. compared the passive drag calculation between a single frontal surface area land-based measure and frontal surface area measures obtained at key events during the stroke cycle of front crawl swimming; however, the frontal surface area was obtained at a very low sampling frequency. In addition, the work of Gatta et al. [18] established frontal area values through the swimming stroke cycle for all strokes, trying to estimate active drag (in all strokes) at different swimming speeds, calculating the surface area 2–4 times lower than the recommended sampling frequency range for swimming motion analysis.

In comparison with the aforementioned studies, the novelty of this paper is in establishing an automatic process to obtain the frontal surface area, improving our ability to collect and analyse more complex and accurate data. The main limitations of this research may be summarized as follows: (a) we only analysed one swimmer for collecting and (b) we analysed in one stroke (crawl). Further studies are warranted to develop a new device for installation in a swimming pool (Figure 17).

## 5. Conclusions

The development of an automated method to obtain the frontal surface area during swimming increases the knowledge of the temporal fluctuation of the frontal surface area in swimming. It would allow the best monitoring of a swimmer in their swimming training sessions.

A novel algorithm for estimating the frontal and lateral area is presented. The computing time allows to obtain a reasonable online representation of the results.

The final objective of this research is to experimentally correlate the linear velocity of the swimmer to the frontal area estimated in this work. For this purpose, a novel device has been designed to follow the swimmer during the activity in order to determine the frontal area and the velocity in a synchronous way. Figure 17 shows the conceptual design of the device.

Our purpose is to experimentally validate model (2) or, if not, to obtain a new model based on experimental results.

## Figures and Tables

**Figure 1 sensors-22-00955-f001:**
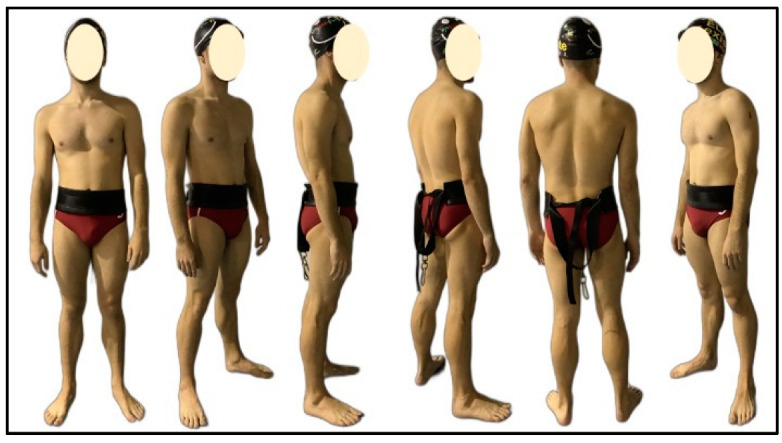
Pictures of the swimmer in different poses.

**Figure 2 sensors-22-00955-f002:**
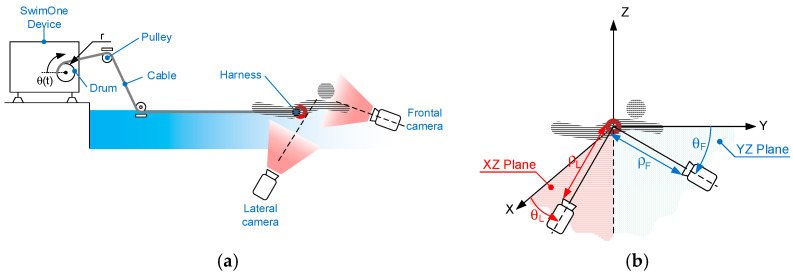
(**a**) Scenario description; (**b**) Cameras position.

**Figure 3 sensors-22-00955-f003:**
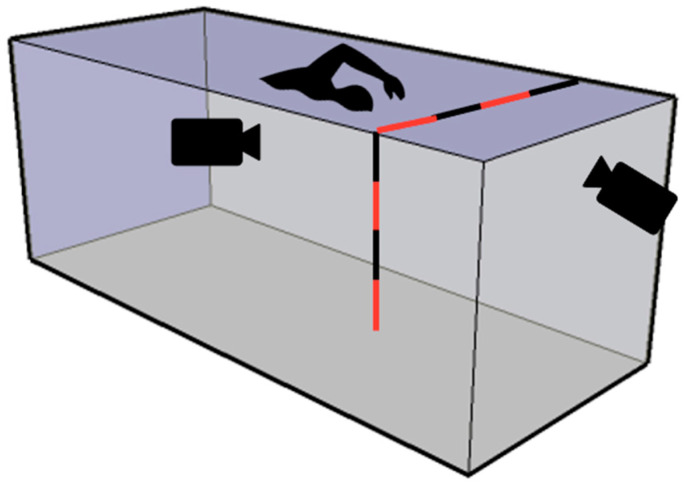
Example of references used during the cameras calibration.

**Figure 4 sensors-22-00955-f004:**
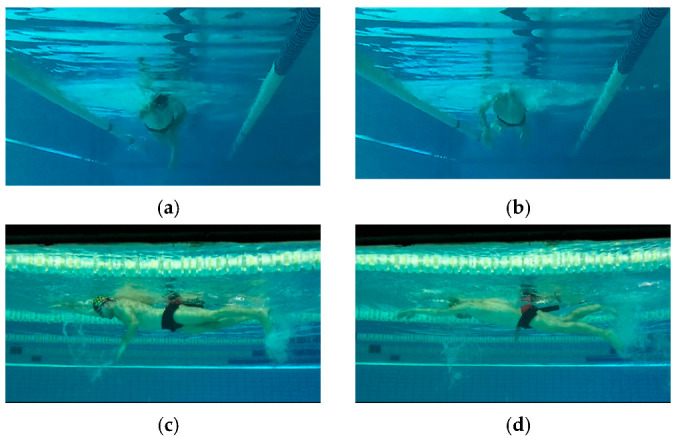
Examples of videoframes obtained by frontal camera (**a**,**b**) and lateral one (**c**,**d**).

**Figure 5 sensors-22-00955-f005:**
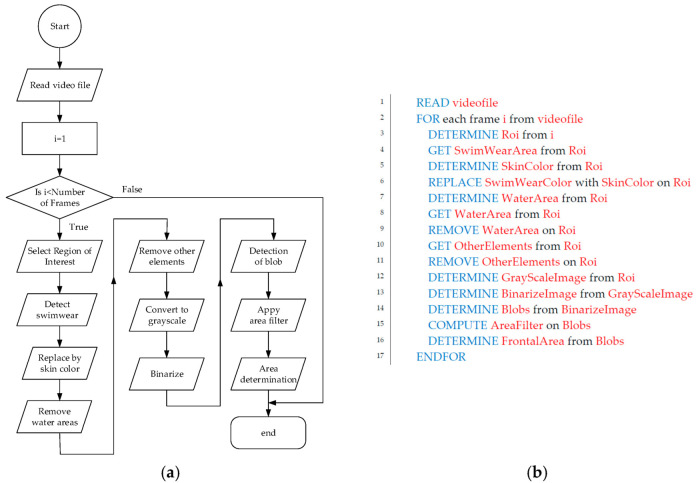
Summary of the algorithm steps for swimmer area determination. (**a**) Data Flow Diagram; (**b**) Pseudocode.

**Figure 6 sensors-22-00955-f006:**
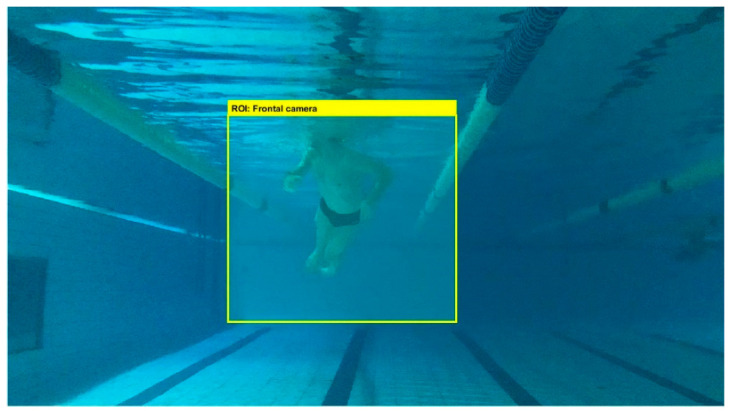
Region of interest (ROI) of the frontal camera.

**Figure 7 sensors-22-00955-f007:**
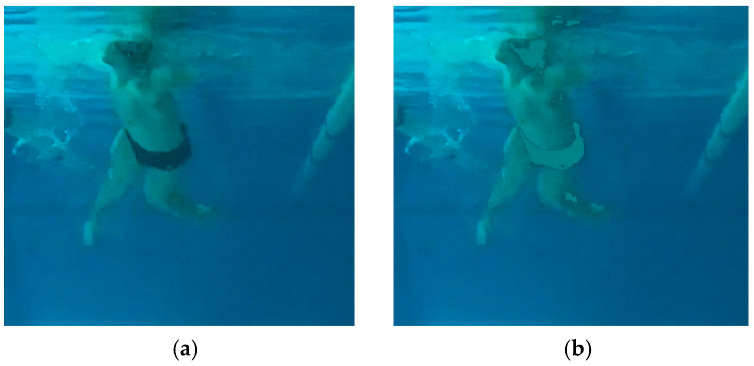
Example of application of the mask to substitute swimwear by skin colour (**a**,**b**) showing pre- and post-processing image, respectively.

**Figure 8 sensors-22-00955-f008:**
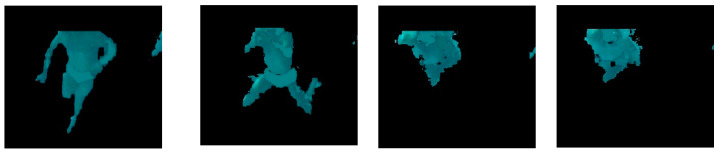
Examples of application of the mask to remove water and elements up to water line.

**Figure 9 sensors-22-00955-f009:**
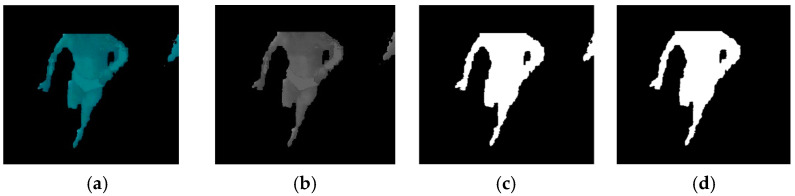
(**a**) Original frame; (**b**) grayscale conversion; (**c**) binarisation; (**d**) area filter.

**Figure 10 sensors-22-00955-f010:**
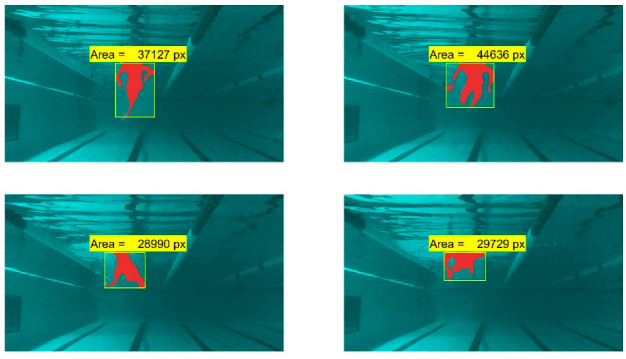
Frontal area estimation of several frames.

**Figure 11 sensors-22-00955-f011:**
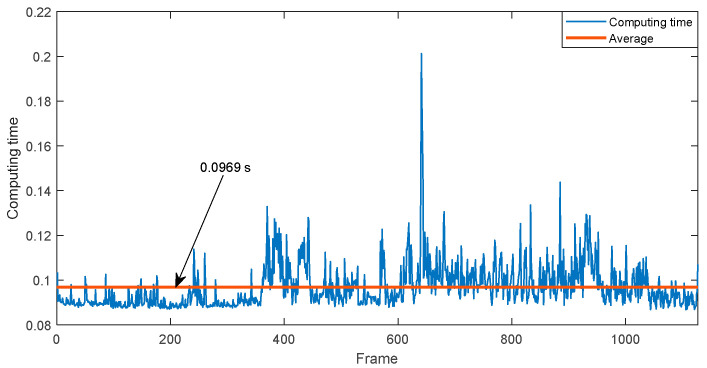
Example of computing time of the frontal area estimation algorithm.

**Figure 12 sensors-22-00955-f012:**
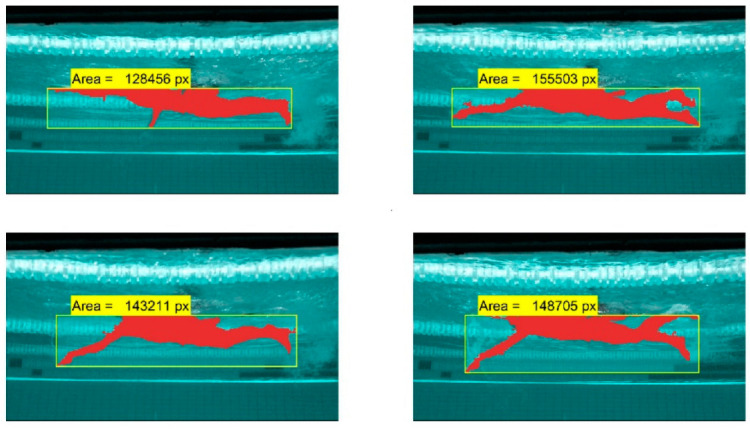
Lateral area estimation of several frames.

**Figure 13 sensors-22-00955-f013:**
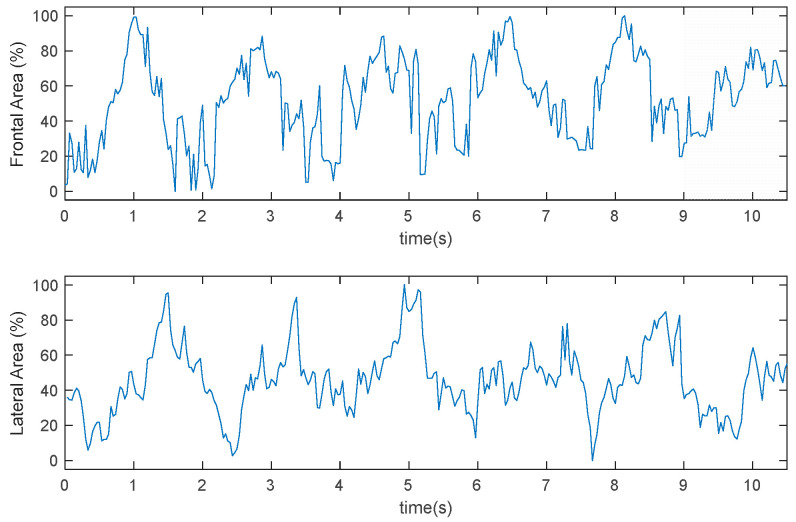
Time evolution of the frontal and lateral area. Test I.

**Figure 14 sensors-22-00955-f014:**
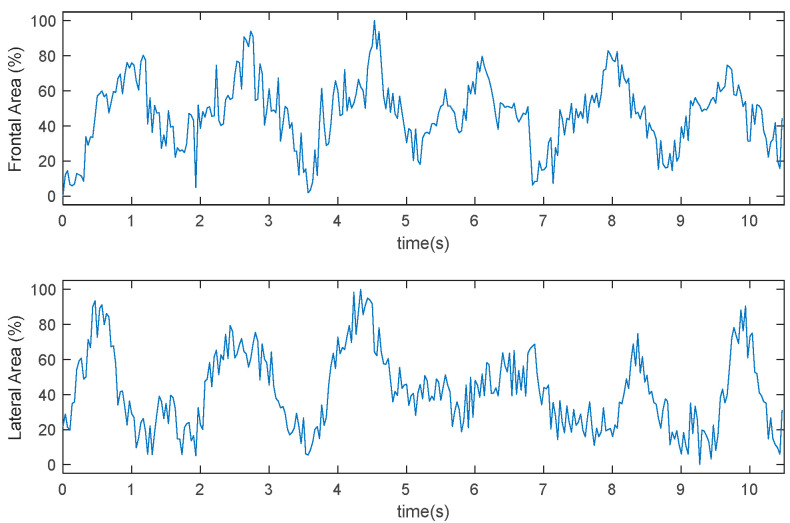
Time evolution of the frontal and lateral area. Test II.

**Figure 15 sensors-22-00955-f015:**
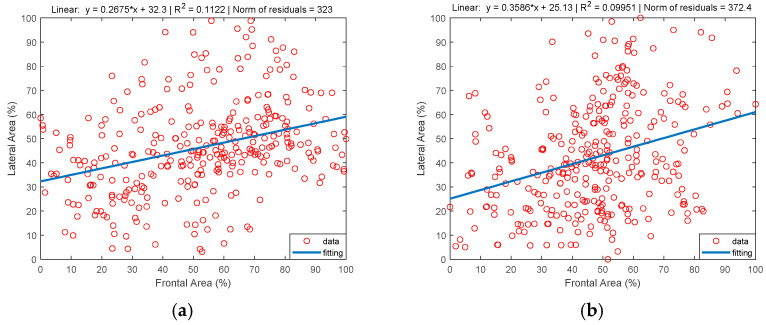
Time evolution of the frontal and lateral area: (**a**) Test I; (**b**) Test II.

**Figure 16 sensors-22-00955-f016:**
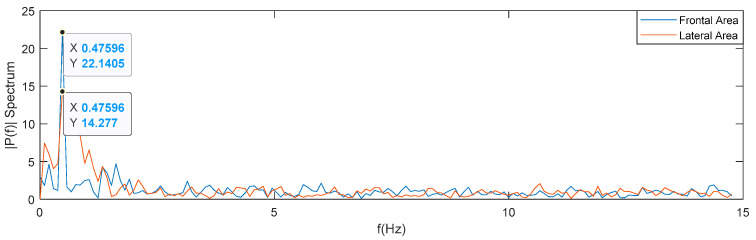
Frequency spectrum of Frontal and Lateral Area for Test I.

**Figure 17 sensors-22-00955-f017:**
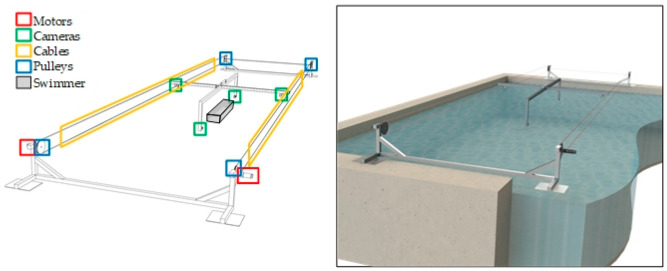
Conceptual device for correlating frontal area to velocity.

## Data Availability

Not applicable.

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
