# Peer review of "Vision-Based System for Automated Estimation of the Frontal Area of Swimmers: Towards the Determination of the Instant Active Drag: A Pilot Study"

_sensors, 2022, doi:10.3390/s22030955_

Round 1

Reviewer 1 Report

The manuscript deals with estimating vision-based frontal area of swimmers. The topic is relevant to the journal and interesting, but there are major issues for revisions.

  • Introduction starts with an equation without "Introduction". I recommend rewriting the Introduction thoroughly. It has no references in the first part.
  • The major contents of Introduction are related work except for the equation. I recommend dividing it into a "related work" section.
  • The core method (2.3. Frontal area detection algorithm) has no algorithms (just a flow chart). I recommend inserting pseudocode for the algorithm and adding sentences on how to determine parameters (L, a, and b).
  • No comparative performance evaluation is presented with state-of-the-art studies.
  • The manuscript is poorly formatted. Figure 14 appeared after Conclusions.
  • More recent references are required.
  • Overall, the manuscript requires extensive revisions to meet the base requirements.

Reviewer 2 Report

This is an interesting paper. There are some points which require clarification:

1) Precisely where are the video cameras located in relation to the water surface? Note that with total internal reflection, an image of the swimmer beneath the water appears above the water inverted. How was this image dismissed from the calculation? The camera angle will also show a different area so the location must be precisely defined.

2) A useful parameter would be the percentage area of the swimmer. This normalised unit would allow a more useful comparison as the swimmer elevates out of the water during the stroke.

3) As the swimmer is stationary, I assume there is no bow wave in front of the swimmer. A bow wave could cause significant interference. 

4) Equation (4). I'm unsure of what is mean by L, a and b. As the swimmer was wearing dark colours, I'm not sure how the swim costume was included in the image.

5) Figure 12 and 13 should include the lateral area also, and some type of correlation between the two areas would be very interesting. 

Minor corrections:

5) Remove all of the "." in the equations. Multiplication is assumed.

6) line 92. what is "ith"?

7) lines 219-220. This sentence does not make sense.

8) line 255. spelling "correlated".

9) line 260. What is "wave" please make this clear.

Round 2

Reviewer 1 Report

The authors revised the manuscript based on the previous review.

Thus, I recommend the manuscript for publication.